# Deep Normed Embeddings for Patient Representation

## Abstract

We introduce a novel contrastive representation learning objective and a training scheme for clinical time series. Specifically, we project high dimensional EHR data to a closed unit ball of low dimension, encoding geometric priors so that the origin represents an idealized *perfect* health state and the Euclidean norm is associated with the patient's mortality risk. Moreover, using septic patients as an example, we show how we could learn to associate the angle between two vectors with the different organ system failures, thereby, learning a compact representation which is indicative of both mortality risk and specific organ failure. We show how the learned embedding can be used for online patient monitoring, can supplement clinicians and improve performance of downstream machine learning tasks. This work was partially motivated from the desire and the need to introduce a systematic way of defining intermediate rewards for reinforcement learning in critical care medicine. Hence, we also show how such a design in terms of the learned embedding can result in qualitatively different policies and value distributions, as compared with using only terminal rewards.

## Introduction

Recently contrastive methods, usually framed as self-supervised learning problems have enjoyed tremendous popularity and success across various domains He et al. (2020); Chen et al. (2020a); Xiao et al. (2021), but their applications for electronic health record data have been limited Yèche et al. (2021). Whilst this can be explained by complexity and noise in medical time series and the difficulty to create medically meaningful augmented versions of the patient states, there is an underlying regularity and structure amongst critically ill patients which we believe can be exploited, to produce a representation using simple geometric priors, working in the semi-supervised [1] setting instead of the fully self-supervised or supervised settings. For this purpose, we introduce a new optimization criteria, using which we embed high dimensional patient states to a lower dimensional unit ball. The embedding has the property that the mortality risk can be associated with the level sphere the embedded vector belongs to, and it can distinguish between variations and similarities between patients states subjected to the same mortality risk, using minimal supervision.

We evaluate our method on a large cohort of septic patients from the MIMIC-III Pollard (2016); Johnson et al. (2016) database. Since our experiments are focused on septic patients, we encode similarly using major systems of organ failure. However, we note that the method can be easily adopted for any subset of patients who exhibit a few major, loosely defined physiological classes of criticality, and can approach higher mortality risks in different ways. By leveraging such basic medical knowledge, our method avoids the need to compute data augmentations to create similar pairs. Unlike in images, augmentations may not produce realistic patient states, due to the high complexity and correlations amongst the data dimensions, and the invariances amongst patient states are less clear. Therefore, we define similarities across two dimensions, a) mortality risk. b) major organ system failures (or a similar notion of similarity), and use a triplet based learning scheme, leveraging local stochastic gradient optimization. We illustrate our method using two simple network architectures a) an auto-regressive GRU network using a fixed horizon, followed by a MLP head -trained on the raw data. b) a single straightforward feed-forward neural network, which uses previous representation

---

[1]Throughout, we use semi-supervised learning to mean learning with some form of partial supervision. We acknowledge that this use may be different from how it may be defined in other work.

learning used in Nanayakkara et al. (2022) [2] The underlying assumptions and geometry which we encode in our training scheme are as follows:

- Each septic[3] patient faces mortality risk, although the underlying physiological causes and infections may be different we can still define a form of similarity using the risk a patient faces. Whilst this can be approached using probabilistic methods, we avoid complications in framing the problem in a probabilistic manner by using semi-supervision. In particular, we require a level set of the unit n-hyperball to consist of the equivalence class of all patient states facing the same risk of death.

- As two patients with the same mortality risk can have two fundamentally physiological causes (for example different organ failures), these embeddings should be on the same level sphere, but on different parts of the sphere.

To achieve these goals, we have to project the embedding into the unit closed ball, in contrast to contrastive methods, where the embedding is constrained to the sphere Chen et al. (2020a); He et al. (2020). Further, we do not have a strict disjoint set of classes, so we cannot use any class based losses such as Deng et al. (2019); Liu et al. (2017). Instead, in addition to similarity in terms of survival, as we stated above we use a softer notion of similarity such as organ failure, noting that it can be possible for a given patient to have multiple organ failures. We also use a triplet based optimization scheme as opposed to using more recent developments in contrastive representation learning such as Van den Oord et al. (2018).

We show several benefits of the proposed method, for both assisting clinicians and for downstream machine learning tasks. For example, the learned embeddings can be used to identify possible new organ failures in advance, and provide early warning signs via the angle of the embeddings and identify increased mortality risk using the norm. The later being considerably better than SOFA score as a predictor for mortality risk for septic patients.

Our work was partially motivated by the desire to introduce a systematic criteria of defining rewards for offline reinforcement learning (RL) applications in medicine. There has been a lot of interest recently in leveraging RL for critical care applications Komorowski et al. (2018); Raghu et al. (2017); Liu et al. (2020). However, there are significant challenges at all levels: a most crucial challenge being a lack of an obvious notion of rewards. Some previous applications of RL for sepsis have for example, have used just terminal rewards Komorowski et al. (2018) (i.e. a reward for the final time point of a patient stay depending on release or death) whilst others have used intermediate rewards based on clinical knowledge and organ failure scores Raghu et al. (2017). Given the limited number of trajectories and the vast heterogeneity amongst critically ill patients, we hypothesize that terminal rewards do not suffice by themselves to learn the desired policies. Indeed, our experiments show that policies and value functions are qualitatively different and more consistent with medical knowledge when we use intermediate rewards. Research in RL has also shown performance and convergence can be improved when the agent is presented a denser reward signal Laud & DeJong (2003). Therefore, we show how a reward can be defined systematically using the learned embeddings, and explore the differences in the policies and value distributions. However, we do keep the RL discussion deliberately brief, and defer a further analysis for future work.

In summary, our major contributions are as follows:

- We propose a novel learning framework where high dimensional electronic health record (EHR) data can be encoded in a closed unit ball so that level spheres represent (equivalence classes of) patients with same mortality risk and patients with different physiological causes are embedded in different parts of the sphere.

- We introduce a loss to encode the desired geometry in the unit ball, since the standard losses in metric learning and contrastive learning were ill-suited for this purpose. Further, we describe a

---

[2]This choice was made to be consistent with state definitions used in the RL step-which in turn was chosen to be consistent with previous research using RL for sepsis.

[3]As we mentioned earlier, we illustrate our method on the specific example of septic patients, but the method is readily applicable with minor modifications for any critically ill patient distribution.

simple sampling scheme suited for this method, and show how the sampling scheme and basic domain knowledge can obviate the need to construct data augmentations.

- We experiment using a diverse sepsis patient cohort, and show how the method can identify mortality risk in advance, as well as identify changes in physiological dynamics in advance.

- We show how this learned embedding can be used to systematically define rewards for RL applications. Such a definition changes the value functions and the policies considerably, when compared with using only terminal rewards.

## Related Work

### Contrastive Learning & Representation Learning for Clinical Time Series

Self supervised learning and contrastive methods have enjoyed increased popularity and success in recent years, particularly in computer vision and natural language applications Chen et al. (2020a); He et al. (2020); Chen et al. (2020b); Deng et al. (2019); Liu et al. (2017); Hoffer & Ailon (2015). Self-supervised learning methods can categorized into two broad categories He et al. (2020). Pretext tasks, where an auxiliary task is solved with the intention of learning a good intermediate representation. Loss function based methods were a representation is learned by directly optimizing an intelligent loss function. We use the latter approach here.

Whilst contrastive representation learning has been popular in other domains, the only similar application to EHR time series we are aware of is Yèche et al. (2021). They propose supervised and self-supervised contrastive learning schemes for EHR data, using a neighborhood criteria for the supervised version.

Our work differs from Yèche et al. (2021) in several ways. First, our method does not require artificial augmentations to define similarity. However we do use very basic medical knowledge about critically ill patients. In that sense, our method belongs to the class of semi-supervised learning rather than self-supervised learning, where most previous contrastive methods were used, with notable exceptions being supervised contrastive learning Khosla et al. (2020) for images, and some recent work on semi-supervised contrastive learning for automatic speech recognition Xiao et al. (2021). This work also significantly differs with respect to the optimization and sampling scheme from all of the previous contrastive methods.

As noted in Yèche et al. (2021), there have been research on using deep representation learning for EHR data both in isolation Lyu et al. (2018), and in the context of RL Killian et al. (2020); Li et al. (2019); Nanayakkara et al. (2022). Lyu et al. (2018) uses sequence to sequence models in both pretext (forecasting future signals) and loss function (autoencoding) contexts. Denoising stacked autoencoders were used by Miotto et al. (2016), to create a time invariant representation of patients. Autoencoders were also used by Landi et al. (2020), to stratify patient trajectories to a lower dimensional vector.

### Reinforcement Learning for Medicine

There has been considerable interest in leveraging RL for medical applications Komorowski et al. (2018); Raghu et al. (2017); Killian et al. (2020); Liu et al. (2020); Raghu et al. (2017); Nanayakkara et al. (2022); Prasad et al. (2020). There have also been guidelines and discussions on challenges associated Gottesman et al. (2019). However, for the best of our knowledge the only other work which deals with systematically defining rewards is Prasad et al. (2020). There, the authors define a class of reward functions for which high-confidence policy improvement is possible. They define a space of reward functions that yield policies that are consistent in performance with the observed data, and the method is general for all offline RL problems. In comparison our method presented here is a simple by product of the learned embedding and has a simple clinical interpretation for critically ill, where reduced mortality is the primary goal.

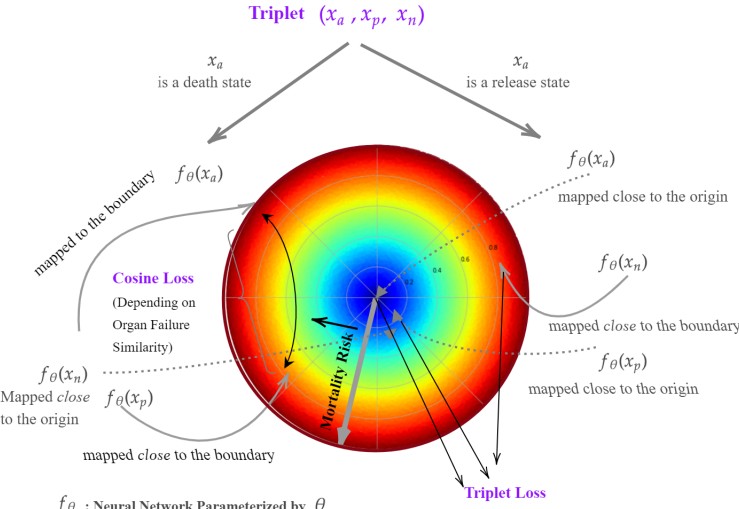

Figure 1: **The proposed training scheme:** We use a triplet based sampling scheme, where 3 patient states are sampled. One of them, the anchor, is always a terminal state (corresponding to death or release), and the others include a near death and a near release state. Our loss function is then defined in terms of the end result of the anchor state as shown in the figure.

## Deep Normed Embeddings: Learning and Optimization

We now motivate our training scheme and optimization criteria, before providing the mathematical formulation. Figure 1 illustrates the geometry we encode on the unit ball, using a 2-dimensional ball as an example. Our optimization algorithm is based on a triplet sampling scheme.

In each triplet the anchor is a terminal state, either a death state or a release (survival) state. The remaining two states are sampled such that, one is a survivor state and the other is a non-survivor state: both in the last $t$ hours of the corresponding stay. (With $t$ being a hyper-parameter, which should be interpreted as being sufficiently *close* to death or release. We used $t = 12, 24, 48, 72$ in our experiments). The state which has the same outcome as the anchor is labeled as positive, the other is labeled as negative. (For example, if the anchor is a death state, then the non-survivor state is labeled as positive and the survivor state as negative. Note that here, the word positive denotes the similarity to the anchor and not the desirability of the given state.)

The triplet of states is then sent through a neural network parameterized by $\theta$, with $f_\theta(x)$ being the lower dimensional embedding of an input $x$ to the network. The optimization scheme *learns* the neural network parameters such that similar states are mapped to proximity while distance between dissimilar states is maximized, and simultaneously the anchor death states are mapped to the boundary and the anchor release states are mapped to *near* [4] the origin. The positive and negative states, are also mapped near the boundary or the origin, depending their end outcome.

In addition to dissimilarity between survival vs non-survival states, we use additional level of dissimilarity among non-survival states that occur due to different organ failure modes. Critically ill patients can face mortality risk in various different ways. For example, septic patients display enormous heterogeneity in the underlying infection and the primary organ failure. Therefore, we require our embedding to identify similarity among the patient states using partial supervision. In our example of septic patients, we use four major organ system scores : i) Cardiovascular ii) Central Nervous System (CNS) iii) Liver and iv) Renal, and pick the organ system with the worst (highest) score as the worst organ system. Each non-survivor state in the triplet

---

[4]The releases states should not be mapped exactly to the origin as even survivors have some risk of mortality, and research has shown there is a substantial readmission risk and a shortened life time for septic patients, even when they survive the ICU stay.

is then annotated with the worse organ system. When the anchor state is a death state, we use a cosine embedding loss, between the two embedded non survivor states. Informally, the goal is to maximize the angle of the embedding of states corresponding to different organ failures and minimizing the angle between two states corresponding to the same organ system failure.

When the anchor is a release state, instead of the cosine embedding loss we use the triplet loss, between anchor, positive and negative. This enables the patient states to be spread across the hyper-ball, and the high mortality risk states to be differentiated from less risky states.

Formally, we optimize the loss function

$$\text{loss}(x; \theta) = \beta(\text{loss}_{\text{terminal}}(x; \theta)) + (1 - \beta)(\text{loss}_{\text{contrastive}}(x; \theta)) + \text{loss}_{\text{intermediate}}(x; \theta) \tag{1}$$

Here $\theta$ denotes the neural network parameters we are optimizing, and $x$ a triplet of the form $(x_a, x_p, x_n)$ (The implementation uses batches of triplets which is the norm in Deep Learning). The loss function in (1) consists of three components: the terminal (or anchor) loss, the contrastive loss, and the intermediate loss for non terminal states. The first two losses are the most important, and are balanced by a hyper-parameter $\beta \in [0, 1]$.

We now describe each component separately. For ease of notation we will use $d(x)$ for $||f_\theta(x)||_2^2$, where $||x||_2$ denotes the $l_2$ euclidean norm on the embedding space. (We use the square of the norm instead of the norm itself purely for the ease of optimization.)

The *terminal loss*,

$$\text{loss}_{\text{terminal}}(x, \theta) = \mathcal{I}_{\{x_a = death\}}((d(x_a) - 1)^2) + \lambda_1 \mathcal{I}_{\{x_a = release\}}(d(x_a)) \tag{2}$$

essentially distributes the terminal states to the correct part of the ball. (with respect to the embedded norm). I.e. the death states are embedded on the boundary and the release states near the origin. As we explained previously we want to be more generous on release states mapped away from the origin, since survivors could exhibit non-trivial mortality risk for critically ill patients. Thus we discount the release term with $\lambda \leq 1$ to encourage the network to learn these patterns automatically.

The *contrastive loss*,

$$\text{loss}_{\text{contrastive}}(x, \theta) = \mathcal{I}_{\{x_a = release\}}\text{tripletloss}(x_a, x_p, x_n) + \mathcal{I}_{\{x_a = death\}}\text{cosineloss}(x_a, x_p, y_{ap}) \tag{3}$$

is responsible for determining the separation of states. This loss depends on whether the chosen anchor is a dead state or a release state. Triplet loss is the standard loss as introduced in Schroff et al. (2015) defined as:

$$\text{tripletloss}(a, p, n) = \max\{||a - p|| - ||a - n|| + \text{margin}, 0\} \tag{4}$$

We used 0.2 for the triplet loss margin. The cosine embedding loss is only considered when the anchor is a death state. This term depends on the similarity of the two non-survivor states $y_{ap}$, where $y_{ap} = 1$ if both the states belong to the same class and 0 otherwise. We experimented with two options for the cosine embedding loss:

(i) The standard cosine embedding loss used in metric learning defined as :

$$\text{cosineloss}(x_a, x_p, y_{ap}) = \begin{cases} 1 - \cos(f_\theta(x_a), f_\theta(x_p)) & y_{ap} = 1 \\ \max(0, \cos(f_\theta(x_a), f_\theta(x_p)) - \text{margin}) & y_{ap} = 0 \end{cases}$$

(ii) Cosine loss based on inner product $<, >$:

$$\text{cosineloss}(x_a, x_p, y_{ap}) = \mathcal{I}_{\{y_{ap}=0\}} < f_\theta(x_a), f_\theta(x_p) >$$

Where $cos(a, b) := <a, b> / \sqrt{<a, a>}\sqrt{<b, b>}$.

Thus, we expect formula (ii) to be similar to (i) near death states, where $\sqrt{<f_\theta(x_a), f_\theta(x_a)>} \approx 1 \approx \sqrt{<f_\theta(x_p), f_\theta(x_p)>}$.[5] Our results in the next sections used the first version with a margin close to 0. Using the second version was more stable in training, but the separation of different organ systems were more clear when the first version was used.

The *intermediate loss* is intended to help the network by mapping near death states near the boundary and near release states near the origin. We note that there are a few hyperparameters in this loss, but our experiments show that the method is quite robust for most *reasonable* hyperparameter choices.

$$
\begin{aligned}
\text{loss}_{\text{intermediate}}&(x, \theta) \\
&= \lambda_2(\mathcal{I}_{\{d(x_p)>1\}}d(x_p) + \mathcal{I}_{\{d(x_n)>1\}}d(x_n)) + \lambda_3(\mathcal{I}_{\{x_a=\text{Death}\}}e^{-\alpha d(x_p)} + \mathcal{I}_{\{x_a=release\}}e^{-\alpha d(x_n)}) \\
&\quad + \lambda_4(\mathcal{I}_{\{x_a=death\}}d(x_n) + \mathcal{I}_{\{x_a=release\}}d(x_a))
\end{aligned}
\tag{5}
$$

This loss comprises of three components. The first term ensures that the embeddings are constrained to the closed unit ball by penalizing if the squared norm of the embedding is greater than one. We noticed that such an implicit regularization is more effective than explicitly constraining the output of the network. The second and third terms help the learning process, by mapping the intermediate (positive and negative) terms close to the boundary or the origin. We use an exponentially decaying loss for the non-survivor states, so the loss only large, if the norm is close to zero. $\alpha$ is a hyper-parameter which chooses the desired decay. Similarly the last term ensures the near release survivor states are mapped close to the origin in general. However we choose $\lambda_4$ to be much smaller than $\lambda_1$ and $\lambda_3$, so that the network can still identify high risk states.

We discuss the effect of hyper-parameter choices, and present an ablation study in the next section (Results). Further, we note that it is also important to use an orthogonal weight initialization Hu et al. (2020) in order to learn a distributed representation on the ball and to prevent dimensionality collapse Jing et al. (2021): This will also be illustrated under Results.

## Results

We will now present some results of our method. The results in this section uses the recurrent neural network architecture. (See supplementary material for implementation details). Some corresponding results for the MLP are presented in the supplementary information.

### Patient Representation on the Unit Ball

For ease of visualization, we present results using representations embedded in the 3d unit ball, however, the method works for embedding into any dimension.

Figures 2a and 2b show histograms of squared norms of the embeddings for all survivor and non survivor patient states (across all time points) in the validation cohort. [6] As the figures clearly demonstrate, the learned embedding associates the norm (or alternatively, the level set $\mathbf{S_k}$ of the form $\mathbf{S_k} = \{x : ||f_\theta(x)||_2 = k\}$) of the embedded vector with mortality risk, with survivor states in general belonging to the lower level sets, and the non-survivor states belonging to the higher level sets. We later show how the norm can be used as an indicator of patient mortality risk, compared with the existing scores such as the SOFA score.

Figure 2c presents a randomly selected sample of patient states, embedded into the 3-dimensional closed unit ball. The colors mark the worst organ system for each state. There is a clear separation amongst different organ failures. We envision, such a presentation can be used to provide real-time visualization to assist clinicians at the ICU. For example, the embedding can be used to identify a patient trajectory heading

---

[5]Note that is this formulation we only use similarity as a loss when the organ failures are different. In either case. the anchor state is a death state.

[6]These results use a network trained with $\beta = 0.75, \lambda_1 = 0.7, \lambda_2 = 10.0, \alpha = 3, \lambda_3 = 0.2, \lambda_4 = 0.05$. Other choices are discussed later.

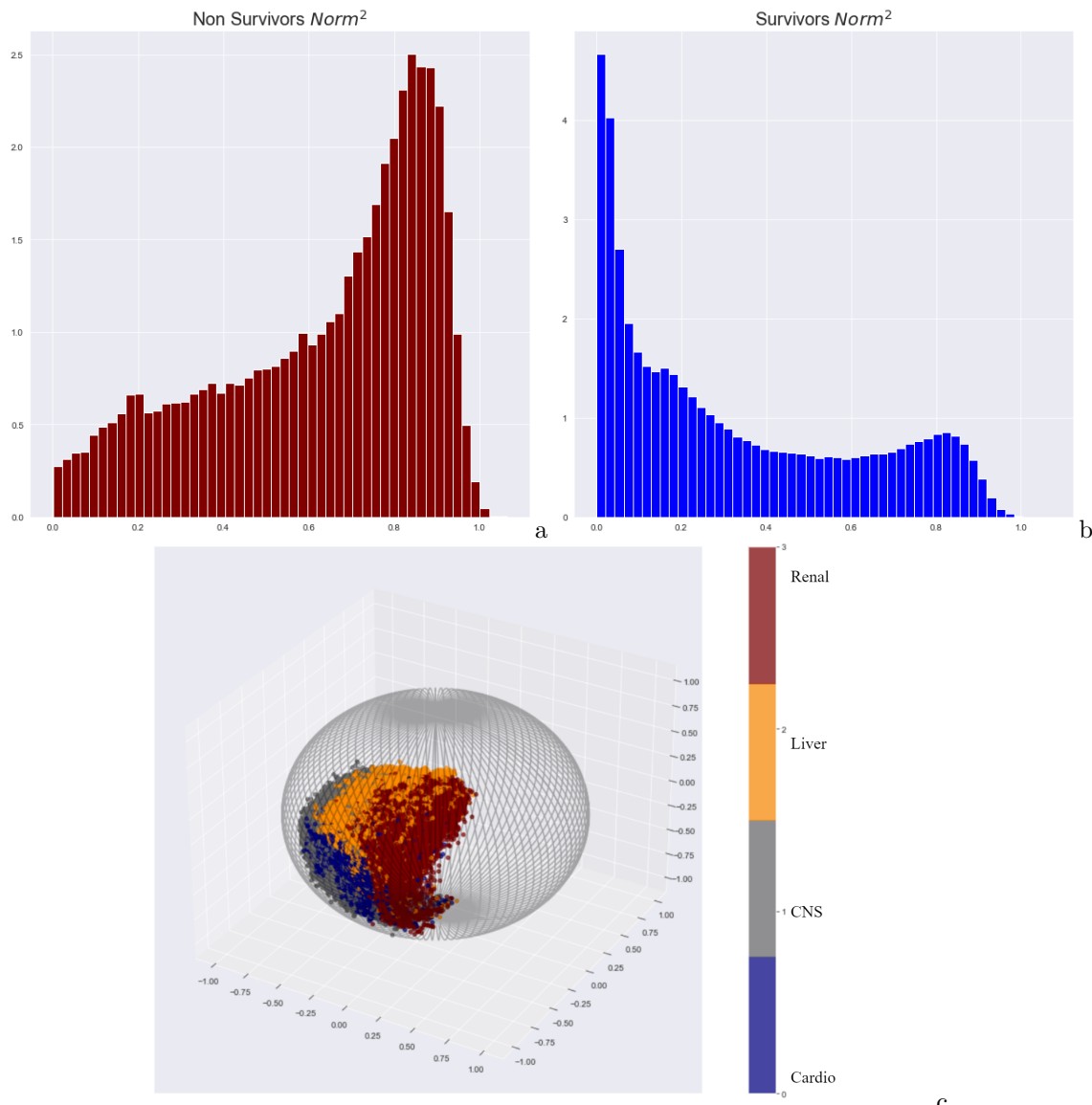

Figure 2: **a:** Norm$^2$ of validation cohort non-survivors, **b:** Norm$^2$ of validation cohort survivors, **c:** A sample of non-survivor patient states, marked by the worst organ system

towards a new organ failure. The embedding being continuous is naturally more granular than the discretized, organ failure scores which were used as an guidance to the network to distinguish different organ failure scores. Indeed, an example of such a patient trajectory is given in Figure 3.

Here, two embedded patient trajectories are plotted in the 3d-unit ball. We focus on the longer trajectory, which is colored in black and green. We focus on the final 50 hrs of this patient's stay. The patient's organ failure scores change at 36 hrs. At this point the patient's the cardiovascular score changes from 4 to 3 and then to 1 at 37 hrs. To show how the embedding *predicts* this change in the underlying physiology before the organ scores reflect it, we color the lines of first 36 hours of the trajectory in black and the last 14 in green. For the first 36 hours the labeled worst organ system is cardiovascular (although, we note for this patient renal and CNS scores were equal to the cardiovascular score.) and hence marked in blue stars. As the cardiovascular score decrease the worst organ system was labeled as CNS and is marked in purple for last 14 hrs. We can notice that the trajectory approaches its final points, even when the organ failure scores do not indicate the increase in cardiovascular scores. Indeed the black lines take the trajectory very close to its end

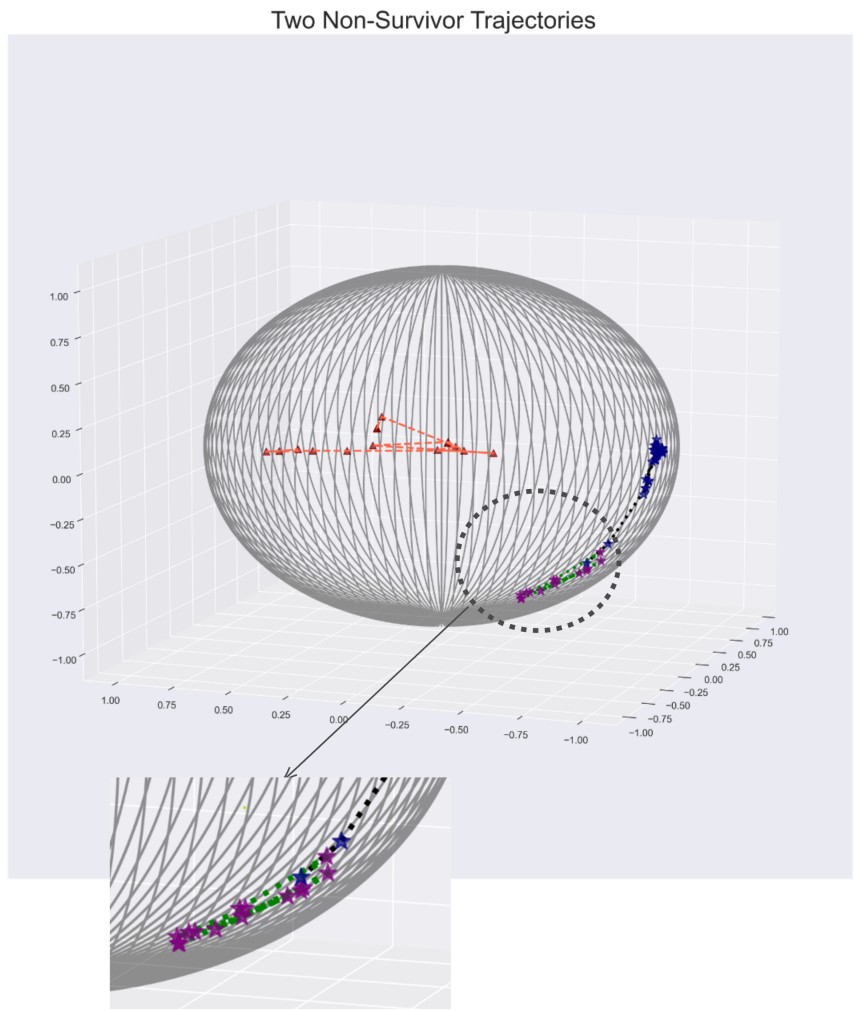

Figure 3: **Embedded trajectories for two non-survivors:** One patient is labeled with star markers and black/green trajectory, the second with triangle markers and orange trajectory. The marker color indicates the system with the highest organ failure score: Cardio (blue), Liver (Maroon) CNS (Purple). The first trajectory is 50 hrs long, black for the first 36 hrs, green for the last 14. The highest severity organ failure changes from cardio to CNS at 36 hrs. The embedding trajectory approaches the cluster a few hours before the organ scores indicate the change (see detail).

set of points. This is an example of how this learned representation can warn clinicians on changes in patient dynamics. As we can see from this example, the representation can identify these patterns from the data and is not constrained by the supervision signal (in this case the organ failure scores) it was given.

The other trajectory is presented for comparison. This is the final 15 hours of another non-survivor. As we can see this patient approaches a different part of the boundary as they become closer to death.

**Hyper-parameter effects**

Now, we will explore the effect of hyper-parameter choices- starting with $\beta$. Figure 4 presents the same patient states shown in Figure 4(c) embedded using different $\beta$ values. Recall that $\beta$ balances the terminal loss and the contrastive loss. The former focus on determining the correct level sphere and the latter on the

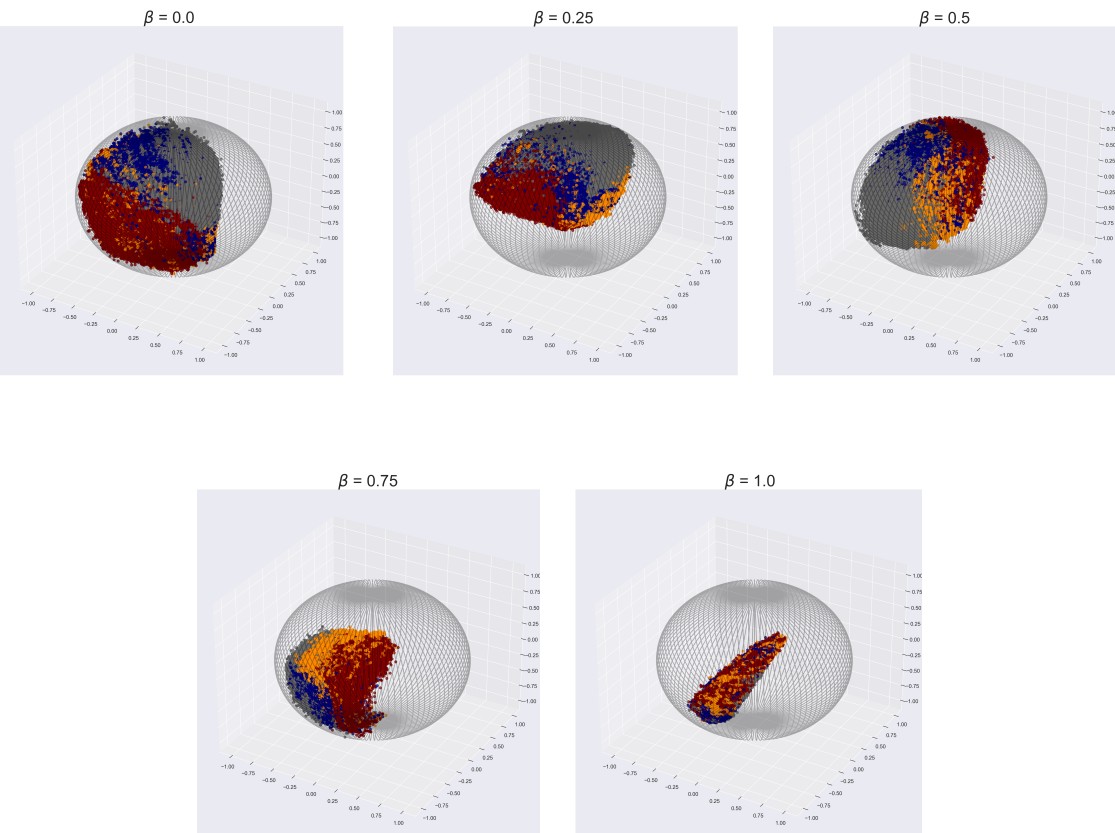

Figure 4: Embedded state distributions for various $\beta$: The labels indicate the worst organ systems as in Figure 2

angle between states. Thus, Figure 4 is not surprising. The states are spread across a larger portion of the ball as $\beta$ gets smaller. The figure also illustrates the importance of the contrastive loss, as when $\beta = 1$ all the states are enclosed into a manifold of much lower volume. The perceptible separation of organ failure modes is also lost. However, encouragingly for all other values of $\beta$, there is a separation.

Figure 5 presents the embedded states, when no orthogonal weight initialization was used. As we can observe, the volume of the space covered is much smaller.

We also examined the effect of $\beta$, on the embedded norm. For this we stratified patient states by: a) survivor and non-survivor, b) times to death or release. Averaged squared norms for different values of $\beta$ are presented in Figure 6. The observations are as expected: higher $\beta$ values on average perform better, in mapping states into a more suitable level sphere. The only exception is for survivor norms, where $\beta = 0.75$ has resulted in lower norms than $\beta = 1.0$. Unsurprisingly, when the terminal loss is not used ($\beta = 0$), the norms between the survivors and non-survivors are similar to each other.

In each case, the averaged squared norms increase with time to death and decrease with time to release.

We then followed the same steps for the intermediate loss. These results can be explored in Figure 7. As expected, when the exponentially decaying terms are excluded (either by setting $\lambda_3 = 0$ or $\alpha = 0$) norms are lower, on average. This was indeed the motivation for using such a term in our loss function. Analogously, when $\lambda_4$ is set to 0, on average norms get larger. It is difficult to evaluate the importance of loss components using Figure 7 in isolation. However, intuitively, there seems to be value in both intermediate loss components.

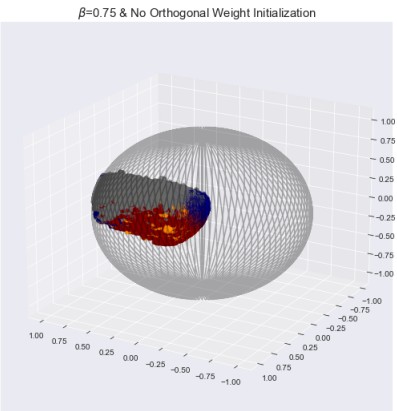

Figure 5: Embedded state distributions without orthogonal weight initialization. The labels indicate the worst organ systems as in Figure 2

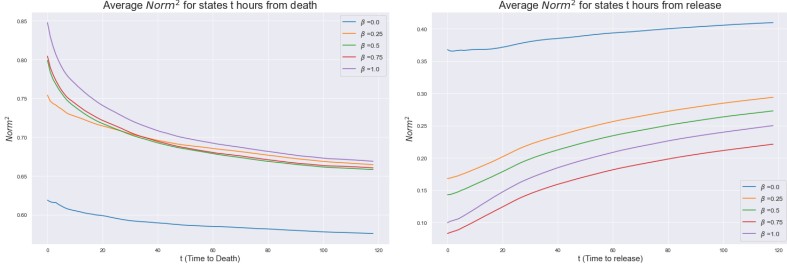

Figure 6: Averaged embedding norm with time to death and release: for different $\beta$

Since the intermediate loss only considers the norm and not the angle, it doesn't have any direct impact on the separation of physiological causes (although it implicitly impacts the effect of $\beta$).

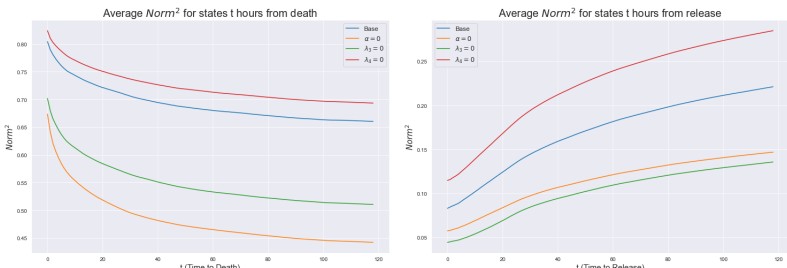

Figure 7: Averaged embedding norm with time to death and release: for intermediate loss choices

We also highly desire some smoothness of the norm trajectory generated by a single patient trajectory. This is particularly important for RL, as we aim to use the difference of the norm (or a monotonic function of the norm) between two consecutive time steps to specify rewards. To quantify this. we computed relative jumps (i.e. $\frac{|d(s_{t+1}) - d(s_t)|}{d(s_t)}$). Averaged results across all patient states are presented in Table 1 for $\beta$ and 2 for intermediate loss.

From Table 1, we can notice that higher $\beta$ values result in more wiggly curves. We also observed this visually. Again, this phenomena can be explained by the form of the loss function. Higher $\beta$ values, focus heavily on the terminal states. Therefore, the loss could be minimized by projecting states closer to the boundary or the origin more frequently.

| $\beta$ | Average relative jump |
|---------|----------------------|
| 0.0 | 0.128 |
| 0.25 | 0.125 |
| 0.5 | 0.162 |
| 0.75 | 0.264 |
| 1.0 | 0.686 |

Table 1: Averaged relative jumps for various $\beta$

| Choice | Average relative jump |
|--------|----------------------|
| Base | 0.264 |
| $\lambda_3 = 0$ | 0.320 |
| $\alpha=0$ | 0.319 |
| $\lambda_4 = 0$ | 0.160 |

Table 2: Averaged relative jumps for various intermediate loss choices, with $\beta = 0.75$

The effects on intermediate loss terms are more interesting. It seems as $\alpha$ (and thus $\lambda_3$) has a smoothing effect. However, $\lambda_4$ seems to be having a increase the magnitude of the jumps.

**Norm as a Predictor of Mortality Risk and Representation Learning for Downstream Machine Learning Tasks**

We investigated how the embedded norm can be used as a predictor of mortality risk. For this, we created auxiliary tasks of predicting if a state is within 12, 24, 48, 72, or 120 hours of death. We further used these tasks to compare the quality of the learned embeddings to other representation learning methods. For the latter goal, we used a linear protocol which is common in the common evaluation protocol in computer vision representation methods He et al. (2020).

First, we calculated the area under the ROC (AUROC), using the norm as the score associated with each state, for each task. For comparison, we followed the same steps with the SOFA score, since SOFA is used as a predictor of mortality for septic patients Ferreira et al. (2001). The results are presented in Table 3. We can notice that the AUROC with respect to the SOFA score is very similar for each task, therefore we also computed the AUROC using a SOFA type, the aggregate score of just 4 organ systems: cardiovascular, CNS, liver, and renal. [7].

| Task (t) | SOFA | SOFA(4) | Norm ($\beta = 0.5$) | Norm ($\beta = 0.75$) |
|----------|------|---------|---------------------|----------------------|
| 12 hrs | 0.717 | 0.746 | 0.847 | 0.836 |
| 24 hrs | 0.716 | 0.741 | 0.828 | 0.8176 |
| 48 hrs | 0.715 | 0.731 | 0.807 | 0.798 |
| 72 hrs | 0.715 | 0.725 | 0.797 | 0.790 |
| 120 hrs | 0.719 | 0.727 | 0.789 | 0.782 |

Table 3: **AUROC for predicting if a state is $t$ hours from death for various $t$.** The statistics are presented first using the full SOFA score, SOFA score just using 4 systems (cardio,cns,liver, renal) (SOFA (4)), Norm with $\beta = 0.5$ and Norm with $\beta = 0.75$

We do note that the SOFA score is an aggregation of different organ failure scores, and a patient can face mortality risk from just a few organ failures. Therefore it is not a perfect score to measure mortality risk. However, it is still used regularly at the ICU to predict mortality risk and it is encouraging that the learned embedding has shown a significant improvement in AUROC. The benefit of our method is that it can indicate the risk *and* the organ failures (or physiological causes in general) responsible. This would not have been possible if the method was approached from probabilistic methods for example.

---

[7]Whereas the full SOFA score uses 6 organ systems

However, we used this problem to investigate the quality of the learned representation, against other standard representation learning methods. For this, we used a linear evaluation protocol by simply fitting a logistic regression model on top of the learned representations. We did this on 100 different train, test splits: training a logistic regression model on one and noting the test AUROC.

For comparison, we learned a recurrent denoising autoencoder Vincent et al. (2008) with the same architecture. Briefly, denoising autoencoders attempt to learn an intelligent representation by reconstructing a corrupted input by a) first projecting into a lower-dimensional space and then b) decoding this embedding. This method is similar to the autoencoding method used in Lyu et al. (2018) [8]. As mentioned under related work, autoencoders are a popular choice for EHR representation learning Miotto et al. (2016); Landi et al. (2020). We noticed that our method significantly outperformed the denoising autoencoder of the same hidden dimension, and thus we also used a denoising autoencoder of a four times larger hidden dimension (12). Further, we used a standard triplet learning scheme. Here, a randomly selected patient state is corrupted by injecting noise to define a positive state. A different patient state belonging to a patient with the opposite end outcome is then selected as the negative state. Then triplet contrastive loss is used as the objective. More implementation details and problem specifications of these methods are included in the supplementary material.

Finally, we also fitted a logistic regression model on the full observations. The results are presented in Table 4. For the normed embeddings we show results corresponding to $\beta = 0.5$ (including a) just the 3d embedding b) embedding and the norm as another feature) which performed the best-numbers corresponding to other choices are stated later. All the models were trained on the same train, test splits.

| Task (t) | Full Observed | Embedding | Embedding + Norm | Denoise Auto (3d) | Denoise Auto (12d) | Triplet |
|----------|---------------|-----------|------------------|-------------------|--------------------|---------|
| 12 hrs   | 0.870         | 0.863     | 0.851            | 0.746             | 0.834              | 0.690   |
| 24 hrs   | 0.847         | 0.837     | 0.829            | 0.726             | 0.811              | 0.681   |
| 48 hrs   | 0.822         | 0.812     | 0.803            | 0.712             | 0.787              | 0.674   |
| 72 hrs   | 0.8097        | 0.802     | 0.795            | 0.704             | 0.778              | 0.674   |
| 120 hrs  | 0.796         | 0.794     | 0.786            | 0.705             | 0.773              | 0.683   |

Table 4: **Test AUROC for predicting if a state is $t$ hours from death for various $t$ : Averaged across 100 train test splits**

As the results indicate, our method significantly outperforms both baselines of the same hidden dimension. In fact, even after increasing the hidden dimension of the next best alternative, its performance was inferior to the method introduced in this work. Indeed, the AUROC of all tasks are quite close to the AUROCs of models trained on the full 27 dimensional raw input. Further, by comparing both Table 3 and Table 4 we can notice that even the one dimensional norm itself is competitive as a risk score even against a model trained on the full input space *specifically* for these tasks.

**Ablation: $\beta$ and intermediate loss**

Next, we will present the AUROC statistics for various $\beta$ values and intermediate loss choices. We present results of fitting logistic regression models across 100 different train and test splits. In addition, we also computed the AUROC using the norm as the score. For simplicity, we averaged over all the above tasks. The results are presented in Table 5 for $\beta$ and 6 for the intermediate loss. [9] It is interesting that with respect to this task $\beta = 0.25, 0.5$ have superior numbers despite higher $\beta$ values focus more on the embedded norm. Even the model trained with no terminal loss, (recall the intermediate loss is still used) performs reasonably.

As Table 6 suggests excluding the exponentially decaying terms (either by setting $\alpha = 0$ or $\lambda_3 = 0$), reduces the AUROC. This observation is consistent the previous ablations (Figure 7). The effect of $\lambda_4$ is however unclear. The norm AUROC reduces, when $\lambda_4 = 0$. However, the logistic regression AUROC improves slightly.

---

[8]However, we use the same simple architecture as above, instead of the attention based architecture used in that work.

[9]Notice that the two tables were generated independently. All the models in the same table used the same train-test splits, however the splits were different across the two tables. Thus, the numbers corresponding to $\beta = 0.75$ in Table 5 and base in Table 5 are different.

| $\beta$ | LR AUROC | Norm AUROC |
|---------|----------|------------|
| 0.0 | 0.783 | 0.693 |
| 0.25 | 0.824 | 0.815 |
| 0.5 | 0.827 | 0.813 |
| 0.75 | 0.818 | 0.805 |
| 1.0 | 0.805 | 0.803 |

Table 5: Averaged (Across splits and tasks) AUROCs for different $\beta$

| Choice | LR AUROC | Norm AUROC |
|--------|----------|------------|
| Base | 0.822 | 0.805 |
| $\lambda_3 = 0$ | 0.810 | 0.800 |
| $\alpha = 0$ | 0.795 | 0.787 |
| $\lambda_4 = 0$ | 0.828 | 0.797 |

Table 6: Averaged (Across splits and tasks) AUROCs for intermediate loss choices- $\beta = 0.75$ in each.

We emphasize that whilst these results are promising, these tasks are artificial. Therefore, performance with respect to the AUROC by itself is certainly not enough to claim that our representation learning method is necessarily superior to other approaches or that a specific hyper parameter combination is superior. Benchmark tasks are popular amongst various machine learning communities. However, evaluating medical machine learning methods (especially unsupervised, representation learning methods) using adhoc tasks can be ineffective and even dangerous. Thus, we intentionally avoided conducting a large number of arbitrary experiments. However, the method introduced here is flexible to be adapted to most similar medical machine learning tasks. Further, it presents enough opportunities to encode domain knowledge.

**Reinforcement Learning: Rewards and Representation**

In this section, we discuss how the learned embeddings can be leveraged for RL. For consistency between RL state spaces and the inputs of the representation learning, the results of this section uses the MLP architecture. In particular, both methods takes the same *state* as input. We present a detailed description of the RL methods and implementation details in the supplementary material.

To be consistent with previous work Nanayakkara et al. (2022), we use deep distributional reinforcement learning using the categorical c51 algorithm Bellemare et al. (2017), which approximates the return distribution with a discrete distribution with fixed support. The state and action spaces are also identical to that work (except when using the embedded vector for state augmentation). We keep the RL methods simple. For example for the results presented here, we do not re-weight the patient distribution when sampling as in Nanayakkara et al. (2022), and we assume actions are taken with respect to the expected value of each value distribution.

As mentioned previously, our intention is purely to illustrate how the proposed low-dimensional embedding can be used to define rewards, aid in state augmentation, and how such a choice affects the recommended policies. Evaluating RL agents in the offline setting is an open problem and an active research area, and the current off-policy evaluations (OPE) are particularly ill-suited for critical care applications Gottesman et al. (2018). Even when OPE methods can be used they are defined in terms of a fixed reward specification, making it impossible to use them for comparison of RL algorithms learned under different reward functions. Therefore, we do not claim the methods proposed here are superior than the existing methods for RL. However, our results show qualitative differences in values and policies that meet clinical intuition, hinting towards the benefit of this formulation.

We experimented with two formulations of intermediate rewards. In each case we used terminal rewards of $-15$ for terminal death states. For terminal survivor states, (release states) we use $15(1 - d(s))$ as the terminal reward. This was done to acknowledge that not all survivors are the same and there could be patients with higher mortality risk even amongst survivors. Indeed medical research have claimed that the

life expectancy reduces significantly even for sepsis survivors. Cuthbertson et al. (2013); Gritte et al. (2021). The scale of 15 was chosen to be consistent with previous work, for example Raghu et al. (2017).

In our first formulation we define, intermediate rewards of the form:

$$r_1(s, a, s') = 0.375(d(s) - d(s'))  \tag{6}$$

[10] where $s, a$ are the current state and action and $s'$ is the next state. Here we use $d(s)$ to denote the square of the norm of the embedded vector of the state $s$. (i.e. $(||f_\theta(s)||)^2$). We have used the current notation for simplicity, noting the slight abuse of notation. This choice has a natural interpretation of minimizing the cumulative increases of risks between consecutive time steps. However, we noticed (by comparing the outputs of bootstrapped networks) that the variance of the learned norm can be high, so $(d(s) - d(s'))$ can only be considered as a noisy estimate of the difference in risk. However, using the boostrapped networks, it is straightforward to include a form of confidence in this estimate, and then consider a regularized reward to reflect parametric uncertainty. We do not do that here do keep our RL presentation brief.

In our second formulation, we defined intermediate rewards using the norm in the same spirit as how SOFA score was used as an intermediate reward in previous work such as Raghu et al. (2017). More specifically, in that work there were two components of intermediate rewards depending on the next state's SOFA score : (i) A change in SOFA score (SOFA score increasing resulting in a negative reward, and decreasing a positive reward) (ii) A negative reward for when the SOFA score does not improve. Further, a 15 or $-15$ terminal reward was given for release or death, respectively.

Therefore, we define intermediate rewards of the form:

$$r_2(s, a, s') = 3.75(d(s) - d(s')) - 0.25\mathcal{I}_{\{d(s')>0.5\}}d(s')  \tag{7}$$

where $s, a$ are the current state and action and $s'$ is the next state. Here we use $d(s)$ to denote the square of the norm of the embedded vector of the state $s$. (i.e. $(||f_\theta(s)||)^2$). We have used the current notation for simplicity, noting the slight abuse of notation.

Notice that in expression of $r_2$ the first term is positive if and only if the norm of the next embedded state is less than the current norm. The second term is a penalty included to discourage keeping a patient at a risky state. For our RL experiments, we used a 10d embedding, and further for the norm calculation we averaged the norms of 10 bootstrapped networks. Both of these choices, were intended to reduce the variance of the estimate. Further, we experimented with augmenting the state representation, with the embedded vector.

Now, we will discuss the changes in the policies. We noticed that when we only use terminal rewards, the percentage of states with no recommended treatment is much higher than with intermediate rewards. This phenomena has been observed in previous research Komorowski et al. (2018). We present a summary of these results in Table 7. Since, there is variability among the treatment recommendations, we present averaged results. The averaging was done using different versions of the function approximating neural network : 5 networks learned independently using bootstrapped patients and weights of the last 3 epochs when the network was trained on the whole training dataset. In addition to averaging the actions recommended by each, we also present results where we average the value functions first and then recommend actions according to the averaged value function. In each, case we can notice that using terminal rewards only causes the action with no recommended treatment more frequently. The breakdown of the full treatment percentages under all 3 reward formulations are presented in the supplementary information.

We will discuss some properties of the value distributions and present the full action distribution in the supplementary material.

## Discussions and Conclusions

In this work, we introduced a novel contrastive representation learning scheme suitable for EHR data. One of the key differences between our method and other constrastive methods, across all application domains, is

---

[10]More generally we can define intermediate rewards of the form $r_2(s, a, s') = \alpha(d(s) - d(s'))$ where $\alpha>0$

| Method | % states with no treatment |
|---|---|
| Clinician | 27.78 |
| Terminal Rewards-Averaged Actions | 52.68 |
| Int. Rewards 1 ($r_1$)- Averaged Actions | 18.33 |
| Int. Rewards 2 ($r_2$)- Averaged Actions | 28.61 |
| Terminal Rewards-Averaged Value Functions | 65.00 |
| Int. Rewards 1 ($r_1$) -Averaged Value Functions | 26.40 |
| Int. Rewards 2 ($r_2$) -Averaged Value Functions | 32.16 |

Table 7: Percentages of states with no treatment

that our method works in the semi-supervised setting rather than purely self-supervised setting. We believe self-supervision using augmentations could be challenging for medical time series, and unfortunately most state of the art constrastive methods depend on heavy augmentations. However, there is enough regularity and domain knowledge which can be exploited, although we do not have strict classes as for example the image domain. Hence, we had to work in the semi-supervised setting rather than a fully supervised setting. Indeed, one of our main aims of this work was to show how minimal and loosely defined supervision and benefit in contrastive learning for clinical applications, and we expect this work to be adapted to reflect different goals in machine learning applications for healthcare.

We have shown that our method has learned to identify mortality risk and changes in patient dynamics in advance in terms of the underlying physiology (via organ failure). We believe such an application can strongly supplement human clinicians at the ICU. The supervision given for this work is minimal and stronger supervision signals about the underlying physiological mechanisms could result in a better and a more interpretable representation. However, this would require more granular data than what is routinely collected at the ICU. Indeed, one of the key challenges in medical time series is that we do not have access to the same quantity or the quality of data as for example natural language.

There has been recent interest in exploring the geometry of deep learning Bronstein et al. (2021). In this work, we use simple geometrical priors using the norm and inner products of a lower dimensional hyper-ball to encode the desired behavior. However, in future work, we plan to explore ways of using stronger geometric priors to encode medical knowledge. We believe such a scheme could also improve interpretability of the representations, as well as improve performance of various machine learning tasks. It is also a potential way to leverage well established mathematical theories of differential geometry and topology (amongst others). However, such a use is far from trivial and would require more research.

We also note that, our method could be improved for task-specific applications through hyperparameter optimization and using different neural network architectures. Our aim was to emphasize on the method and the associated geometric intuitions, and thus we did not focus on finding the *optimal* hyper-parameters. Similarly, we note that the performance could be improved by using recent advances in contrastive learning such as what is introduced in Chen et al. (2020a); He et al. (2020).

Finally, we showed how the learned embedding can be used to define rewards for RL and how as a result the distribution of values and the policies change considerably. Whilst we have only used the norm of the embedding for RL results presented here, we anticipate this method can be used in other ways for RL and control. For example, the organ system changes can be considered if we can define rewards in terms of the inner product of two consecutive vectors. However it is not immediate how this should be done, so we defer this to future work. We may also interpret the lower dimensional embedding as an action induced patient trajectory and a simplified dynamic patient model (where the action conditioned dynamics will have to be estimated). This should allow us to use model based control methods, and the low dimension could enable us to use more traditional control methods. However this too, would require more research and is another direction we want to explore. Unfortunately, *all* RL methods in medicine are subjected to challenges at all levels, including evaluation. Therefore, we do not make any claims about the performance of the learned RL policies, rather we emphasize the method and how it can be used to set up the RL framework, more systematically compared to previous work.

## Broader Impact Concerns

We emphasize our aim of this work, is to introduce a novel, *potentially* impactful computational approach. However, as with all computational and data driven approaches to medicine, significant human evaluation is necessary before such approaches can be utilized at the ICU. Thus, we certainly don't recommend this method for practical deployment at its current state.

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

## Supplementary Information

## A    Data sources and preprocessing

We used a fixed cohort for all our experiments. This cohort consisted of adult patients ($\geq 17$) who satisfied the Sepsis 3 criteria Johnson et al. (2018) from the Multi-parameter Intelligent Monitoring in Intensive Care (MIMIC-III v1.4) database Johnson et al. (2016); Pollard (2016). The excluded patients included patients with more than 25% missing values (of vitals and scores) after creating hourly trajectories, patients with no weight measurements recorded and patients discharged from the ICU but ended up dying a few days or weeks later at the hospital.

We used already pivoted, hourly vitals and scores available through the MIMIC-project. However, labs were measured more infrequently-in most cases once in every 8-12 hours. Therefore the lab values were imputed using a last value carried forward scheme, with the interpretation that the recorded data is the last measured lab. For both vitals and scores, missing values using a last value carried forward scheme.

More specifically the state consisted of :

- **Demographics**: Age, Gender, Weight.

- **Vitals**: Heart Rate, Systolic Blood Pressure, Diastolic Blood Pressure, Mean Arterial Blood Pressure, Temperature, SpO2, Respiratory Rate.

- **Scores:** 24 hour based scores of, SOFA, Liver, Renal, CNS, Cardiovascular

- **Labs:** Anion Gap, Bicarbonate, Creatinine, Chloride, Glucose, Hematocrit, Hemoglobin, Platelet, Potassium, Sodium, BUN, WBC.

For RL and for the MLP based representation learning we also used the representation learning used in Nanayakkara et al. (2022). These states included 4 cardiovascular states and a 10 dimensional lab history representation.

For RL, we used the same action definitions as Nanayakkara et al. (2022). For fluids, this was the total hourly volume of fluids (adjusted for tonicity). However for vasopressors it was the maximum norepinephrine equivalent hourly dose (mcg/kg). The vasopressor 1/2 cut off was 0.15 mcg/kg/min norepinephrine equivalent rate. The corresponding cutoff for fluids was 500 ml for fluids. Action 0 denotes no treatment.

In summary, the markov decision process (MDP) used for RL is:

- **State**: The 41-dimensional state space described above

- **Actions**: A 9 dimensional discrete action space, where vasopressors and fluids can take values $0, 1, 2$. 0 indicates that treatment wasn't administrated.

- **Rewards**: Several choices were used (see main text).

- **Time Step:** 1 hr

## B    Reinforcement Learning

In this section, we will briefly mention some RL background.

RL is a framework for optimizing sequential decision making. RL can be formalized using a Markov Decision Process (MDP), consisting of a 5-tuple $(S, A, r, \gamma, p)$. This includes state and action spaces $\mathcal{S}, \mathcal{A}$, a (typically unknown) Markov probability kernel $p(|s, a)$, which gives the dynamics of the next state, given the current state and the action and a reward process with a kernel $r(|s, a)$. A policy $\pi$ is a possibly random mapping from states to actions.

Given a discount factor $\gamma$, the return is defined as the cumulative discounted rewards : $\sum_{t=1}^{\infty} \gamma^t r_t$, which is a random variable. The objective of an RL agent is to optimize some functional of the return, usually its expected value (Induced by a policy and environment dynamics).

Thus, the *value function*, $V^\pi(s) = \mathbb{E}_{p,\pi}[\Sigma_t \gamma^t r_t(s_t, a_t) | s_0 = s, \pi], \forall s \in S$, is defined as the expected future discounted rewards when following policy $\pi$ and starting from the state $s$. The *Q-function*, $Q^\pi(s,a) = \mathbb{E}_{p,\pi}[\Sigma_t \gamma^t r_t(s_t, a_t) | s_0 = s, \pi, a_0 = a], \forall s \in S, a \in A$, which returns the expected future reward when choosing action $a$ in state $s$, and then following policy $\pi$.

Distributional RL methods, attempt to learn the entire probability distribution of the return, rather than focusing on the expected value. Therefore distributional methods can be used to define actions with respect to criteria different from the expected value.

## C  Implementation Details

### Contrastive Representation Learning

For the autoregressive model, we only considered states after at least 12 hours from admission. For these states, we first send the past 12 hr history (up to and including the current time), through a GRU based recurrent neural network. Then, we concatenated the current GRU hidden state with the current observations and send the new input through a MLP head. The final layer is sent through a tanh non-linearity.

We trained all our networks for just 10 epochs (passes through the training data). In each case, we monitored a validation loss (with respect to the same loss that is optimized) and saved the weights of the network, corresponding to the minimum validation loss.

We used standard, mini-batch stochastic gradient based optimization using Adam Kingma & Ba (2014) with a batch size of 128 and a learning rate of $3 \times 10^{-5}$. In sampling batches, we first sampled a number of patients equal to the batch size and their respective terminal states were taken as the anchor states. Then for each patient, a non-survivor state and a survivor state (in the last $t$ hours) from two different patients were drawn randomly and depending on the end outcome of the anchor state, these states were labeled as positive or negative. The worst organ scores corresponding to each state, were also noted.

We further used a weighted sampling scheme, where non-survivors were sampled more frequently (as the anchor). However, this was purely due to the heavily imbalanced nature of our cohort where around 90% of the patients were survivors.

The following table lists all the hyper-parameters used in our implementation. Note that we have mentioned $\beta$

For MLP, the same contrastive loss hyper-parameters were used. However, we used larger batch sizes of size 256, It had 12 hidden layers of 512 hidden units and ELU non-linearities. The optimization details were the same as above.

### Baseline Representation Learning

**Denoising Autoencoder**: We used the architecture described above as our encoding neural network. However, we did not use the tanh non-linearity at the end. During, training we injected noise by randomly zeroing out entries with a probability of 0.1. The decoder was a MLP with one hidden layer of 128 dimension and a ELU non-linearity. We used a batch size of 128 and Adam as the optimizer with a learning rate of $3 \times 10^{-5}$. This network was trained for 25 full epochs, and we used the weights of the network with the best test loss (computed with corruption).

**Triplet Contrastive Learning**: For the triplet contrastive method, we again used the same architecture, except for the tanh non-linearity. We normalized the output so that all outputs are unit vectors.

We trained by first randomly selecting a patient, and then a state. This was the anchor. We then, injected independent Gaussian noise to each dimension to create a positive version. Next, we sampled a patient that

| Hyper-parameter | Value |
|---|---|
| RNN layers | 2 |
| MLP layers | 8 |
| RNN hidden dimension | 128 |
| MLP hidden dimension | 512 |
| MLP activation functions | ELU |
| Weight Initialization | Orthogonal |
| Optimizer | Adam |
| Learning rate | $3 \times 10^{-5}$ |
| Batch size | 128 |
| Non-Survivor Sampling weight | 5 |
| $\alpha$ | 3 |
| $\lambda_1$ | 0.7 |
| $\lambda_2$ | 10 |
| $\lambda_3$ | 0.2 |
| $\lambda_4$ | 0.05 |

Table 8: Contrastive learning hyper-parameters

had a different terminal outcome. A random time point of this second patient was taken as the negative state. We again, used batch sizes of 128 and Adam with the same learning rate. The triplet loss margin was 0.2

**Auxiliary Tasks**

For logistic regression models, we created 100 different train and test splits by first, sampling 80% of patients out of the total cohort, and then taking the data-points of these patients as the training data and the rest as test data. For a given set of features, we trained a logistic regression model on each of the training data and evaluated the test AUROC.

**Reinforcement Learning**

For RL, we used the c51 distributional algorithm Bellemare et al. (2017), with a 51 dimensional support, using batch sizes of 100 and Adam as the optimizer. For bootstrapped networks, we first generated a random number between $k$ 0.6 and 0.85, and selected a random $k\%$ sample of patients. Then, the network was trained on these patient trajectories.

Other relevant hyper-parameters are noted in the following table.

| Hyper-Parameter | Value |
|---|---|
| Support size | 51 |
| Maximum value | 18 |
| Minimum value | -18 |
| $\gamma$ | 0.999 |
| Batch size | 100 |
| Optimizer | Adam |
| Learning rate | $3 \times 10^{-4}$ |
| $\tau$ | 0.005 |

Table 9: RL algorithm hyper-parameters

# D More Results

In this section, we briefly present results when the MLP architecture was used. However, note that these results were not generated using the same patient cohort as 2.

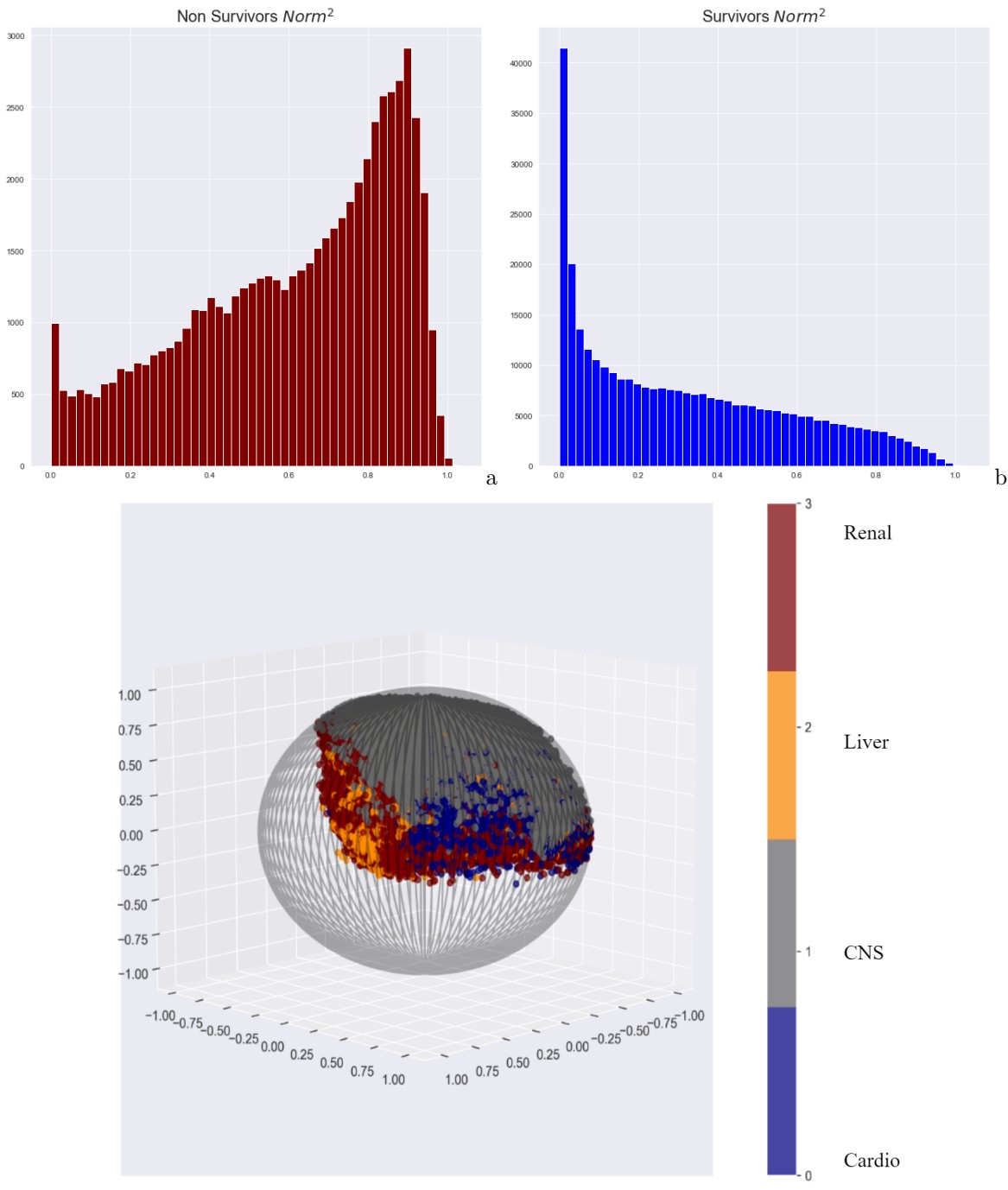

Figure 8: **Results of the MLP model a:** Norm$^2$ of validation cohort non-survivors, **b:** Norm$^2$ of validation cohort survivors, **c:** A sample of non-survivor patient states, marked by the worst organ system

## E    More RL & Control : Results and Discussions

In this section, we briefly discuss some additional results of leveraging our method for RL and control.

For better comparison, we present a table (Table 10) with the percentages of all actions across the whole cohort under the 3 reward schemes. The results presented here are derived from the averaged value distributions, using bootstrapped ensembles.

| Action | Terminal Rewards. | Int. Rewards 1 ($r_1$). | Int. Rewards 2 ($r_2$) | Clinician |
|--------|-------------------|-------------------------|------------------------|-----------|
| Vaso 0 Fluids 0 | 65 | 26.4 | 32.2 | 27.8 |
| Vaso 0 Fluids 1 | 19 | 11.7 | 0.03 | 23.7 |
| Vaso 0 Fluids 2 | 9 | 18.3 | 58.2 | 31.8 |
| Vaso 1 Fluids 0 | 2.8 | 7.6 | 0.9 | 1.2 |
| Vaso 1 Fluids 1 | 3.3 | 23.5 | 1.1 | 3.2 |
| Vaso 1 Fluids 2 | 0 | 0.2 | 0.1 | 4.0 |
| Vaso 1 Fluids 0 | 0.04 | 12 | 7 | 1.2 |
| Vaso 2 Fluids 1 | 0.02 | 0.03 | 0.2 | 2.5 |
| Vaso 2 Fluids 2 | 0.02 | 0 | 0.01 | 4.4 |

Table 10: Percentage of recommended actions under different schemes and the clinician

We note that there is a considerable difference between recommendations among the reward schemes. Evaluating between different policies using historical data is one of the hardest challenges faced by any application of RL or control to medicine. Therefore, we don't claim any specific scheme is necessarily better at this point.

However as we have mentioned previously the first formulation does have a natural meaning for critically ill patients, and its increased vasopressor recommendation is consistent with previous RL work for sepsis Nanayakkara et al. (2022), and recent medical research Shi et al. (2020). We suspect the reasons for the second reward choice to recommend less vasopressors could be that the clinicians usually prescribe vasopressors for high risk patients, thus there are less high risk patient states with no vasopressors administered in our observed data. ($r_1$ penalizes staying at a high risk state by $-0.25d(s')$) This could potentially be addressed by using offline RL methods for minimizing the effects of distribution shift, but such efforts are differed for future work.

Now, we will compare the optimal values under different formulations.

We will present our results using three different formulations: (i) using only terminal rewards, (ii) using only terminal rewards but augmenting the state with the embedded vector, (iii) using intermediate rewards (No embedded state augmentation). Each was trained using the same hyper-parameters for 8 epochs.

Since the value itself is defined in terms of the reward choice, we scaled all the values using a minimum, maximum scaling scheme, so that for each formulation the values fall in the interval $[0, 1]$. We then, explored the differences of values amongst survivors and non-survivors, expecting a noticeable difference at least when the states are close (in time) to their eventual final outcome.

Due to the more pronounced difference, we will present results which use $r_2$, first.

Figure 9 presents box plots of scaled optimal values of patient states. In this figure the intermediate rewards use the formulation $r_2$. An analogous figure, with $r_2$ can be found in the supplementary material. For each, we present the box plots for all survivor states, all non-survivor states, survivors states within 24 hours of release, and non-survivor states within 24 hours of death. It is interesting to note that, the differences in values are most perceptible when intermediate rewards are used. This makes more sense clinically, than results when only terminal rewards were used, where the median non-survivor values are high even when they are 24 hours from death. Moreover, there seem to be a slight increase in difference between survivor and non-survivor quartiles, when the representation learning is used. This is especially noticeable in the last 24 hours of each set of patients.

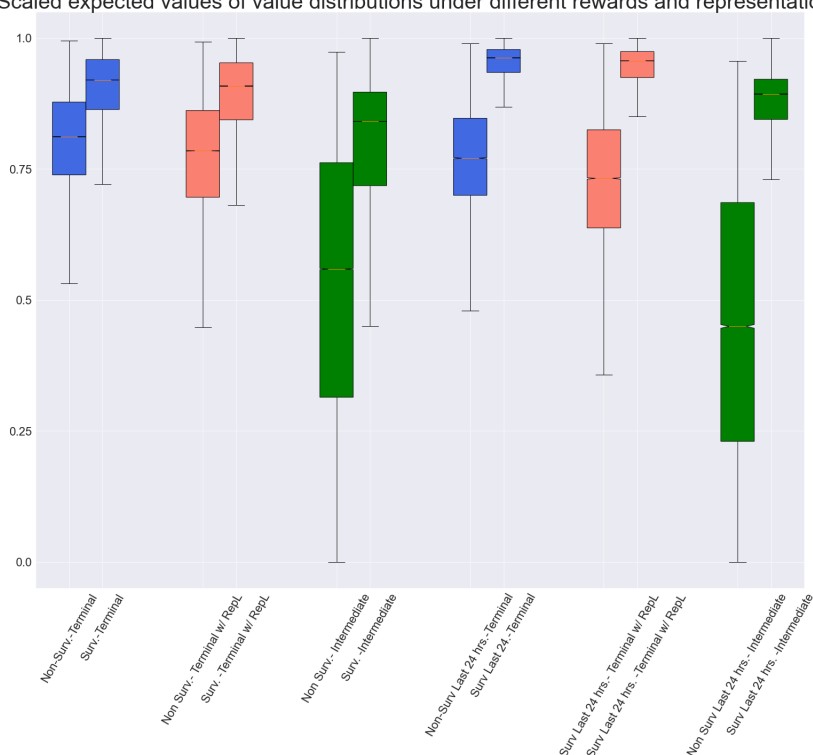

Figure 9: **Box plots of optimal values :**   The results are shown for different reward schemes and representations.

Figure 10 presents box plots for optimal values for all 3 reward choices. We can notice that when $r_1$ is used instead of $r_2$ the differences between survivor and non-survivor values are less pronounced. However, there are still interesting differences when compared with terminal rewards. For example, variance and interquartile range of survivors are much higher. (Recall that the values are scaled using a min-max scheme) In addition, the values of survivors are no longer concentrated near 1.

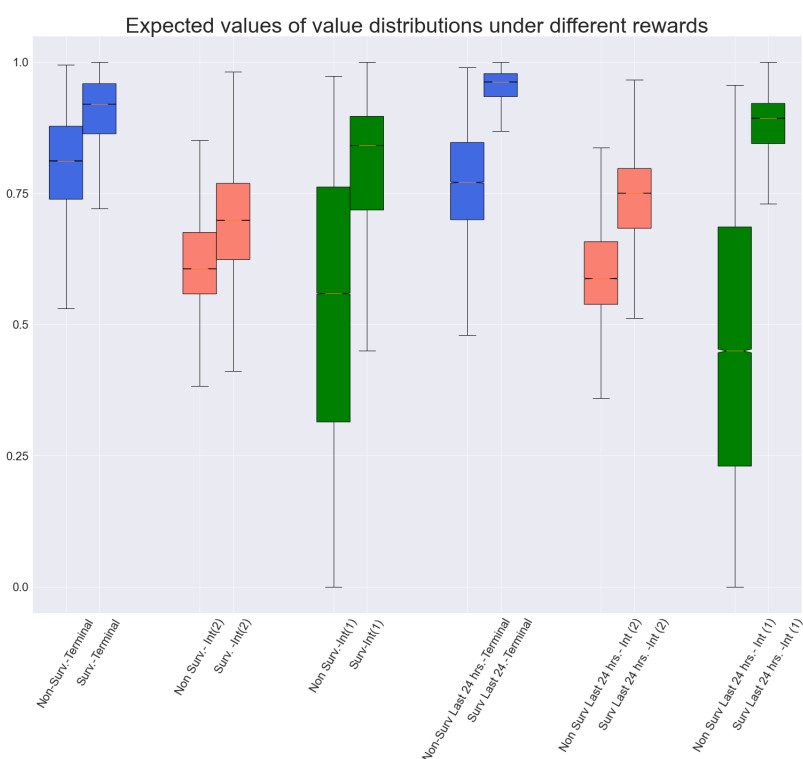

Figure 10: **Box plots of optimal values:** The results are shown for different reward schemes

