# OpenReview forum: "Deep Normed Embeddings for Patient Representation"
_TMLR — Rejected by TMLR_

### Review · Reviewer_Kf4E · 2022-06-02

**Summary Of Contributions:**

This paper studies the representation learning on EHR data with semi-supervised contrastive learning. A novel method is proposed to train an encoder that projects the EHR data to a closed unit ball. The norm represents the mortality risk and the direction represents the cause of mortality. In particular, the learned representation can be used for defining rewards in reinforcement learning. The proposed method is evaluated on mortality prediction tasks in MIMIC-III for septic patients.

**Broader Impact Concerns:**

No specific concern for broader impact.

**Requested Changes:**

1. Cosineloss equation typo: $y_p$ should be $x_p$
2. Make a formal definition of the task and evaluation metrics (important)
3. Ablation study for different parts of the loss function (important)
4. Comparison with other baselines (e.g. predicted probability from a classifier). (important)


**Strengths And Weaknesses:**

Strengths:
1. This paper researches an interesting topic that uses RL to recommend treatment.
2. The combination of contrastive learning with supervision is interesting.

Weakness:
1. The rationale of forcing the norm of healthy patient embeddings to be zero is not justified. I don’t see the benefit of making embeddings different healthy patients collapse to the origin. Also, computing the cosine distance (inner product) between two vectors with different norms can be biased.
2. The goal of training the representation is to compute the distance $d(s)$ of state $s$ which is a value in $[0,1]$. In fact, the $d(s)$ is what really matters in the downstream RL task, not representation itself. The estimation of d(s) can be achieved by many alternatives. For example, a simple classifier with a Sigmoid function. However, there is no comparison with other baseline methods. Especially, since the embedding learned from this method is not outperforming random forests with original features in Table 1, why not define the calibrated prediction of RF as $d(s)$?
3. The design of loss is super complicated. It consists of 10 parts and 7 hyperparameters. There is no ablation study showing which part is essential.
4. As a study following the setting of Nanayaskkara et al. (2022), there is no clue that supports that the proposed representation learning method is superior to the previous work. In fact, the importance scores in Table 2, reported with respect to the dimension of embeddings are less interpretable than the importance scores on exact variables reported by Nanayaskkara et al. (2022).
5. The formulation of the RL problem is unclear. Some key parts are missing, including the definition of the task, the objective function, and the metric of evaluation. Although there are some citations to the previous paper, it is still essential to describe these essential elements in this paper. This is also connected to the previous point because this problem makes it unclear whether the proposed method is better.

---

> ### Author Response · Authors · 2022-06-20
> **Response to  Reviewer Kf4E**
>
>
> We thank Reviewer Kf4E for their time and helpful comments.
>
> We have made several revisions to address the concerns of all reviewers. In particular, we used an autoregressive model on the raw data that does not depend on previous representation learning methods and restructured the mortality prediction task. We also considered several baseline representation learning methods and evaluated them using a linear classification protocol. Further, we included several ablation studies.
> We will upload the revised version of the article on 06/20/2021-with a detailed description of the revisions.
>
> Please note our responses to your specific concerns next.
>
> $\textbf{Weaknesses and revisions}$
>
> 1) The reason for making the origin an idealized perfect health state was to make the norm a monotonically increasing function of the underlying patient risk. Whilst, the origin can of course be replaced by any other arbitrary point by translating the unit ball, this would make the optimization more involved, and would not prevent the collapsing of healthy states. In theory, we could attempt to learn an arbitrary manifold representing patient states, but then we would lose the geometric properties that were used in the optimization scheme. One particular possible design would be to let the $\textit{healthy manifold}$ be a closed sphere and enclose the patient states in an annulus, but then we would have to compute a distance between a point and this set to compute the increased risk.
>
>  There was a typographical error in how the cosine loss was presented (only in the equation) in the original paper, this is now corrected. The inner products are taken in the embedded space, with respect to the Euclidean norm.
>
> 2) We agree that it $\textit{may}$ be possible to use a probabilistic classifier as $d(s)$, however as we mentioned in the paper, it’s ambiguous how the classification problem should be formalized. For example, if we use the probability of death within 48 hours as $d(s)$, maximizing $\textit{cumulative}$ probabilities of death in 48 hrs from all states, doesn’t make sense as a clinical goal.  The benefit of a semi-supervised method is that it may be possible to identify an appropriate $\textit{physiologic}$ risk by weak supervision. However, we acknowledge that our loss function only attempts to make the $norm^2$ only a monotonically increasing function of such an idealized concept of risk. We aim to explore ways of improving this in future work.
>
>  3) As per your requested changes, we included an ablation study of the hyper-parameters. We focused on the parameter $\beta$ and the intermediate loss components.
>
> 4) As we mentioned previously, we reproduced the results without any previous representation learning. We also noticed that using the linear evaluation protocol, the current embeddings significantly outperformed the cardiovascular states of Nanayakkara et al. (2022). However, it is important to note that the cardiovascular states have a number of benefits beyond improving downstream machine learning tasks. Therefore, we expect the current method to work together rather than in place of these representations.
>
> As a brief comment on our previous results: Importance scores which were presented in that table were computed for all of the state components, which included the previous representation methods described in Nanayakkara et al. (2022). Whilst, we acknowledge that this was not strong evidence of the current method being superior it gave some indication.
>
> 5) We included formulation details of both RL and the supervised learning problem in the supplementary information.
>
> Regarding points on evaluation metrics for RL, we deliberately excluded this because currently there are no suitable methods of evaluating the quality of treatment strategies learned by RL, using historical data. As we mentioned in the paper, using importance sampling-based OPE methods are unreliable and potentially dangerous to evaluate RL policies (Because policies that recommend no treatment even to high-risk states can be preferred). This makes comparing two representation learning methods or two reward formulations almost impossible.
>
> Thus, we compared our method using several other baselines using the supervised learning problem. However, we did not change the RL section for the above reasons.
>
> We believe we have addressed all your concerns and implemented all of the requested changes.

---

### Review · Reviewer_g6pG · 2022-06-04

**Summary Of Contributions:**

In this paper, the authors introduce a method for learning representations of patient's Electronic Health Records (EHR) data, and apply it for mortality prediction amongst sepsis patients. In summary, their contributions are:
1. A new contrastive representation learning objective to encode the desired geometry of the patient representations. In particular, they project the EHR data to a closed unit ball, where the origin represents the "perfect" health state, the Euclidean norm represents the mortality risk, and the boundary represents mortality.
2. The authors mention that the work is also partially motivated by defining intermediate rewards for RL problems, as in previous work (Raghu, 2020). Hence they use the intermediate representations from (1) to define the intermediate rewards for the RL task.

**Broader Impact Concerns:**

This is not included. Authors can discuss ethical implications of relying on predictive ML models in practice, and the potential of learning biased representations.

**Requested Changes:**

Addressing all of the weaknesses above is critical to securing a recommendation for acceptance. Some specific changes:
1. Providing additional (stronger) baselines for learning EHR representations.

2. Hyperparameter tuning of proposed approach and baselines.

3. Including confidence intervals and additional metrics.

4. Evaluating the method on at least two one more supervised learning task.

5. Ablation study for the contrastive loss.

6. Improving the quality and consistency of the writing of the paper (please see above). I only provided a few examples, but the paper needs to be thoroughly revised, and long sentences should be minimized.

7. There's a lot of repetition in the introduction (between what they propose, what they did, and what the contributions are). I would suggest that they make it more concise.

8. They compare their approach on page 2 to "previous methods, where the embedding is constrained to the sphere", but provide no citations. Can you add the citations and elaborate on what was proposed in this earlier work that you are referring to?

9. Elaborate what's happening in all figures and tables more precisely. For example, in Figure 1, authors mention that the loss is defined as shown in the figure, but that's not explicitly shown. Also style of math in figures and text should match.

10. Making the code open-access

11. Clarifying in detail implementation details, such as sampling frequency of non-survivors.

**Strengths And Weaknesses:**

Strengths:
1. The authors introduce a new formulation for contrastive learning for learning representations of EHR data.
2. The authors show the merit of the approach for two tasks: (a) supervised mortality risk prediction (b) reinforcement learning for treatment recommendation.

Weaknesses:
1. While the new approach is promising, it has two main weaknesses. First, it relies on the use of latent representations introduced by previous work (Nanayakkara, 2022). Therefore, it adds another step to the pipeline. How would it work with processing raw data instead? Second, while the formulation is novel, the improvement in performance does not seem to be significant. The best results are achieved only when the learned representations are combined with the raw representations and then fed into a random forest. Otherwise, using a simple LR model outperforms the learned representations.

2. The authors mention several hyper parameter choices, and claim that they did not search for optimal parameters. This makes it difficult to reach any conclusions. Some hyper parameter values are also missing.

3. The analysis for both the supervised task and the RL task is brief, and the latter is acknowledged by the authors. The results are not thorough. The authors can provide additional metrics and confidence intervals for the first task. They should also consider performance across different patient subgroups since they include age and gender (which should be rephrased as sex if it's female/male).

4. The authors cite a couple of work but they don't successfully contextualize theirs in the context of what's already out there.

5. They also do not have strong baselines that aim to learn EHR representations, nor for the intermediate reward function.

6. The mathematical notation is a bit sloppy. All the inputs are vectors, and that should be clarified. Dimensions should be explicitly defined. All the losses should be explicitly and clearly defined. Is $cos(x_a, x_p)$ supposed to measure the cosine of the angle?

7. The authors do not conduct any ablation studies to investigate the impact of incorporating different loss terms. This is crucial.

8. In the abstract, authors should be more specific about their contributions and include some numbers. They say that they introduce a training scheme, but I can't see that apart from the loss. That being said, a lot of implementation details are missing, such as batch size, learning rates, bootstrapping, etc. They mention implementation details in different sections, but it should all be brought together and improved for reproducibility.

9. The presentation of the paper is poor. There are several writing issues in the paper. There are run-on sentences, such as footnote 2 on page 1. Capitalization of words is used sporadically, such as "Reinforcement Learning" in the abstract, and "Mortality Risk" on page 1. Abbreviations need to be introduced at the first instance, like "E.H.R." in the abstract. In other places the authors use EHR. Additionally, authors constantly refer to what they previously stated as "as we stated previously" or "as mentioned previously". Instead they should restate what they previously mentioned to the author. There are also spelling mistakes like in footnote 4, no instead of not. The manuscript also needs restructuring, such as moving methods from end of the manuscript to the methodology section. The introduction also repeats ideas all across and lacks focus. Finally, for the most part the citation style is incorrect. For example, in text citations that do not explicitly use the author as a noun should be in the form of (Author, year). There are other writing issues across the paper like the first two lines after equation 2 on page 5.

10. No mention of open-access code.

11. Not all claims are supported by empirical evidence. For example, they say that orthogonal weight initialization performed best.

12. Supplementary material are not referenced properly.

13. The method should be evaluated for other supervised learning task. MIMIC-III has three tasks that may be of interest: mortality, length of stay prediction, decompensation. The authors should discuss how it could be expanded considering the incorporation of organ failure in the semi supervised task.

14. Authors should discuss how the method could consider combinations of organ failure, not just the one with the highest score.

15. Authors mention learning curves, they should show a few examples.

---

> ### Author Response · Authors · 2022-06-20
> **Response to Reviewer g6pG**
>
> We thank Reviewer g6PG for their time and comments.
>
> We have made several revisions to address the concerns of all reviewers. In particular, we used an autoregressive model on the raw data that does not depend on previous representation learning methods and restructured the mortality prediction task. We also considered several baseline representation learning methods and evaluated them using a linear classification protocol. Further, we included several ablation studies.
> We will upload the revised version of the article on 06/20/2021-with a detailed description of the revisions.
>
> $\textbf{Weaknesses and Revisions}$
>
> Note that the numbers denote the numbering of the weaknesses. Due to word constraints, we will focus on the main points. We hope the rest will be reflected in the revisions.
>
> 1) We reproduced the results, exclusively using the raw data. We used previous representation learning methods (Nanayakkara (2022)  purely for consistency with state definitions of RL (Which in turn was chosen to be consistent with other RL efforts). However, we do acknowledge that it's not the best choice for the current paper, and thus the results were replaced by the latter method. We still used the previous representation learning for RL- so that there is consistency between inputs to each network (value, and reward).
>
> When restricting the use to raw data, however, we used an autoregressive RNN method (with a fixed short horizon) before an MLP head- instead of using exclusively a feedforward network. This was to capture some short-term temporal patterns. More details are discussed in the paper.
>
> We restructured our presentation of the mortality prediction task. However, we will first comment on the previously included results.
> A LR model learned $\textit{specifically}$ for a given task, on the $\textit{full raw data}$, including representation learning of (Nanayakkara (2022) did outperform the norm in AUROC. However, the representation cannot be compared with a model. Rather, the learned representation can only be compared with another representation. The norm represents a more general notion of $\textit{risk}$ than a probability of dying within a specific time frame and is a one-dimensional score, which summarizes data of much a larger dimension.  The fidelity of a learned representation can only be compared either as a dimensionality reduction method or as the improvement on the performance when it is used.
>
> To evaluate the learned representation, we borrowed the linear protocol from computer vision representation learning methods. Briefly, we fitted LR models on top of the learned representations along with the full raw data, and other baseline methods and compared the AUROC averaged across 100 train and test splits.
>
>
> 2) We included detailed sections, which discuss the effect of hyper-parameters.
>
> 3) We restructured and expanded on the supervised learning problem (see point 1). However, note that our main goal of this work is to introduce a method to supplement clinicians. Therefore, we believe presenting minute details on the performance of one specific task such as subgroup metrics is irrelevant and dilutes the theme of the paper. Further, focusing on artificial performance measures can be dangerous in medical applications. Thus, our goals were to propose a general method that can be adapted to different clinical problems and to show how the learned representations could assist downstream machine learning tasks. We believe our presentation provides sufficient proof for both of these claims.
>
> 4) We have expanded the background and related work section.
> 5) We included several baselines for EHR representations As for the intermediate reward function, to the best of our knowledge there are no accepted clinically meaningful reward choices. Further, it’s impossible to compare the performance of different reward choices. Thus we deliberately excluded such comparisons.
> 6) We defined the losses and fixed some typographical errors.
>
> 7) We included a detailed ablation study. (See also 2))
>
> 8) We have restructured the implementation details section. This section can now be found in the supplementary material. We have also mentioned these experiments in the source code which will be released after anonymity is no longer required.
>
> 9) See the revised paper.
>
> 10) The code will be open-sourced. We intentionally avoided directing reviewers to the code to predict anonymity: As suggested by TMLR.
>
> 11) Fixed.
>
> 12) See revised paper.
>
> 13) We did not evaluate the method on another supervised learning task.  However, we expanded on the previous task significantly. See also (3).
>
> 14) The method can be trivially modified to any function discrete of organ failure scores (or indeed any $f : \mathbb{S} \to \{1,....N}$ where $N$ is an integer). The only necessary change is that $y_{ap}$ will now consider the discrete $N$ labels of two different states.

---

### Review · Reviewer_Ei7V · 2022-06-06

**Summary Of Contributions:**

The paper proposes a representation of patient time-series data, particularly in the ICU motivated by the need to i) represent mortality risk and ii) notion of similarity. Since the choice of disease is sepsis (occuring due to potential organ failures after infection), authors choose to represent the patient state in a closed unit norm ball, where the level sets of equal norm of the representation indicates similar mortality risk, and the region of the unit ball corresponds to specific type of organ failure. A "semi-supervised" contrastive objective is formulated to achieve these goals where a Neural Network is a function mapping from the patient state space to an embedded space with above properties. The objective consists of multiple parts i) Terminal loss that encourages the mortality risk to correlate to the norm of the embedding function. ii) Contrastive loss that encourages higher similarity based on patients who present similar organ risk failures and encourage dissimilarity if not, and iii) an intermediate loss function for regularizing the embeddings themselves.

**Broader Impact Concerns:**

I suggest simply adding a short Broader Impact statement to indicate that is is prototype ML and health work and is not recommended for practical deployment.

Unclear whether anonymized patient ID should be up in the paper, but is useful for reproducibility. I suggest moving it to the supplement.

I do not have any other concerns.

**Requested Changes:**

1. Discussion on intermediate loss
2. Justification of concerns asked regarding Figure 2 and 3
3. Experiments on other auxiliary choices such as imputation methods and effect on representation quality and effects on downstream tasks.
4. More clarity on choice of reward function for downstream tasks.
5. I would have ideally liked to see at least one more time-series health dataset like eICU to replicate experiments in some form (maybe not sepsis but another task). So that the generalizability of the proposed cost function can be determined. I am saying this since the authors already claim that the formulation is general enough.

**Strengths And Weaknesses:**

Figure 1: Add legend of what $\theta$ is.

Describe/Define the triplet loss.

Figure 2:"Non survivor norm" - The slight bump at 0 is weird. Do the authors know the problem here? The dip in the end is also not clearly interpretable to me.

Figure 2: "Survivors norm" - This figure is also unclear as I am not sure the data is so clear that there is a clear decreasing trend over the norm. I would like to understand why this could be justified. I feel like this points to potential concerns with the design of the intermediate loss (exponential terms?) and these are not well justified in their current form. Is there a clinical motivation to use thsoe? And how does that choice influence the representations. This seems to warrant more discussion.

Final concern on Figure 2(c) - What is the interpretation of the global location of the different diseases and the fact that there is a large region of the unit ball that is not covered. Does this imply something about whether the maximization of the dissimilarity (between patients dying at different potential organ failures) is a very sensitive optimization problem or is it indicative of something really physiologically meaningful (which I doubt). I am raising this concern because it then seems like the optimization could be sensitive. And the three state representations chosen may not be enough to learn the (dis) similarity. Would like to understand more of what's happening here.

Can the authors include a lot more background on how and when different aspects of the state are collected? For instance, when Lab measurements are collected, like BUN, WBC etc, is there a way to know why these were ordered? Or is that part of the ICU protocol? I can imagine sampling frequencies being quite different for parts of these state representations. In that case, how do authors deal with missing data? What is the impact of different imputation methods on the quality of the representations?

Figure 3: Is the trajectory plotted on a test patient or a training patient?

In the RL tasks, the definition of rewards seems to not include the action at all, however there usually are intentional clinical constraints on which interventions make sense, and are particularly a function of what the set of potential organ failures are being encountered. Can the authors add those constraints easily to the reward function? I think it would make their results better and in a better position to actually interpret policies (though I appreciate that the authors do not want to as it is highly challenging to make general claims).

---

> ### Author Response · Authors · 2022-06-20
> **Response to Reviewer Ei7V**
>
> We thank Reviewer Ei7V for their time and helpful comments.
>
> We have made several revisions to address the concerns of all reviewers. In particular, we used an autoregressive model on the raw data that does not depend on previous representation learning methods and restructured the mortality prediction task. We also considered several baseline representation learning methods and evaluated them using a linear classification protocol. Further, we included several ablation studies.
>
> We will upload the revised version of the article on 06/20/2021-with a detailed description of the revisions.
>
> Please note our responses to your specific concerns next.
>
> $\textbf{Strengths And Weaknesses:}$
>
> 1) We defined the triplet loss and added a legend for $\theta$.
>
> 2) Regarding your concerns regarding Figure 2: We agree that these should have been addressed in more detail. Thus, we included several results including averaged norms with times to death and release for several hyper-parameters combinations.
>  Some of the behavior you mentioned is no longer visible in our latest results and we hypothesize that the slightly counter-intuitive patterns in the histograms are due to the optimization dynamics and model failures. For example, the bump near 0 for non-survivors could be due to the network incorrectly identifying some non-survivor states as very healthy states. We also noticed that there was a tendency for the model to pick one of the two extremes (close to 0 or 1) for some states- especially for higher values of $\beta$. Indeed, this was the motivation for including the intermediate loss. The exponentially decaying term was in particular, included to discourage mapping non-survivor states near the origin. Similarly, the dip at the end is also likely to be an optimization artifact. The loss is high only when the embedded norm of a risky state is mapped away from the boundary.
> We included several ablation studies which provide better insight into the roles of each loss component. We hope these results will address your concerns further.
>
> 3) The optimization process was indeed very sensitive and the global distribution is determined by optimization dynamics. The revised paper presents this global distribution of states for different values of $\beta$. As we can notice for small $\beta$ values, a larger part of the ball is covered, however, as the focus on the norm increases the space becomes smaller. However, we currently do not have any interpretation of the absolute position of the states. We have made minor edits to the manuscript to discuss these observations and limitations.
>
> 4) We used pivoted data presented by the MIMIC project hosted on GCP Big Query (With manual verifications). Vitals and scores were measured hourly, and as you mentioned labs were measured infrequently (typically once every 8-12 hrs). We used the last value carried forward scheme for labs, with the interpretation that the value is the last recorded value. The same method was also used whenever vitals and scores were missing (as long as the missingness of a patient was less than 25%. This information is presented in the supplementary material in the revised paper.
> Unfortunately, we are not aware if the data describes how the labs were ordered. (However, we will have to reverify this).
>
> 5) This patient is a training patient for the revised model. (We used the same patient trajectory for consistency, however, the second patient trajectory was changed)
>
> 6) This is an excellent suggestion, but it is difficult to encode such constraints to the reward function- at least without considerable research. Indeed, our primary motivation in proposing this method was to eliminate the need of designing rewards manually. The hope of RL is that by a proper objective function (which is defined through rewards) such constraints will be learned automatically using experience. However, as we also mentioned in the paper, we acknowledge that this is incredibly difficult in medicine.
>
>
> Revisions
>
> 1, 2, and 4 were discussed above.
> 3) We also included a section, we discuss the effect of a number of such choices. Whilst, imputation methods were not considered. We did consider the effect of $\beta$, intermediate loss, weight initialization, and contrastive loss.
> Assuming you meant the reward function for RL, we hope the above discussion address these points. We also expanded this section of the paper.
>
> 5) We would have ideally liked to have done this, and evaluated our method on a variety of other data sources and problems. However, we had to focus on several other experiments which were needed for ablation studies and representation learning baselines.
>
> We included a Broader Impact Statement following your suggestions. We also strongly agree that this work is currently at a very initial state and not ready for deployment.

---

### Decision · Action_Editors · 2022-07-17

**Recommendation:** Reject

**Comment:**

Dear authors,

I regret to reject this paper. Although this must be disappointing, I think the quality of reviews in this case was high. My recommendation is that you should take into account the comments of the reviewers to the best of your ability. I believe doing so will greatly increase the chances of this paper being accepted.

Summary of justifications provided by the reviewers:
* The authors did not fully respond to the suggested comments, especially: (i) hyper parameter tuning such as learning rate and (ii) the presentation of the paper still needs significant work, (iii) code could be anonymously shared using existing tools such as: https://anonymous.4open.science (iv) there are a couple of unclear points, such as defining t as a hyper parameter and using it to define the timeframe of the prediction task.
* The authors did not add an additional health task. I am a bit concerned now of the generalizability of the method, although there is no concern on the correctness front. I also feel authors could significantly improve presentation of results, the formatting etc. in order to improve readability. The paper could benefit from some edits to the presentation as well. Please add confidence intervals to the results.